# Retinal Organoids from an AIPL1 CRISPR/Cas9 Knockout Cell Line Successfully Recapitulate the Molecular Features of LCA4 Disease

**DOI:** 10.3390/ijms24065912

**Published:** 2023-03-21

**Authors:** Pedro R. L. Perdigão, Bethany Ollington, Hali Sai, Amy Leung, Almudena Sacristan-Reviriego, Jacqueline van der Spuy

**Affiliations:** Institute of Ophthalmology, University College London, London EC1V 9EL, UK; pedroperdigao@cnc.uc.pt (P.R.L.P.); b.ollington@ucl.ac.uk (B.O.); h.sai@ucl.ac.uk (H.S.); amy.w.s.leung@ucl.ac.uk (A.L.); a.reviriego@ucl.ac.uk (A.S.-R.)

**Keywords:** AIPL1, CRISPR/Cas9 gene editing, inherited retinal dystrophy, Leber congenital amaurosis, retinal organoid

## Abstract

Aryl hydrocarbon receptor-interacting protein-like 1 (AIPL1) is expressed in photoreceptors where it facilitates the assembly of phosphodiesterase 6 (PDE6) which hydrolyses cGMP within the phototransduction cascade. Genetic variations in *AIPL1* cause type 4 Leber congenital amaurosis (LCA4), which presents as rapid loss of vision in early childhood. Limited in vitro LCA4 models are available, and these rely on patient-derived cells harbouring patient-specific *AIPL1* mutations. While valuable, the use and scalability of individual patient-derived LCA4 models may be limited by ethical considerations, access to patient samples and prohibitive costs. To model the functional consequences of patient-independent *AIPL1* mutations, CRISPR/Cas9 was implemented to produce an isogenic induced pluripotent stem cell line harbouring a frameshift mutation in the first exon of *AIPL1*. Retinal organoids were generated using these cells, which retained *AIPL1* gene transcription, but AIPL1 protein was undetectable. AIPL1 knockout resulted in a decrease in rod photoreceptor-specific PDE6α and β, and increased cGMP levels, suggesting downstream dysregulation of the phototransduction cascade. The retinal model described here provides a novel platform to assess functional consequences of AIPL1 silencing and measure the rescue of molecular features by potential therapeutic approaches targeting mutation-independent pathogenesis.

## 1. Introduction

Leber congenital amaurosis (LCA) is an autosomal recessive retinopathy which leads to rapid loss of vision in early childhood and typically presents as very poor vision, nystagmus, and reduced or absent full-field electroretinograms (ERG) [1]. Causative genetic mutations of LCA have thus far been identified in 25 genes that are typically expressed in the retina or retinal pigment epithelium [2].

Aryl hydrocarbon receptor-interacting protein-like 1 (AIPL1) is expressed in pineal and photoreceptor cells [3,4] and mutations in the *AIPL1* gene are associated with LCA type 4 (LCA4) [3]. AIPL1 plays a key role as a co-chaperone alongside HSP70 and HSP90 to facilitate the assembly of photoreceptor phosphodiesterase 6 (PDE6) [5,6,7,8]. PDE6 hydrolyses cyclic GMP (cGMP), which is vital for light-mediated photoreceptor hyperpolarisation within the phototransduction cascade [9]. The loss of *Aipl1* in knockout mice results in a reduction in PDE6 subunits and subsequent accumulation of cGMP, leading to rapid photoreceptor cell death [5,7,10,11,12].

The *AIPL1* gene is 41 kb, comprising six exons, and is found on chromosome 17 (Genome Data Viewer https://www.ncbi.nlm.nih.gov/genome/gdv/, accessed on 20 June 2022). The AIPL1 protein contains 384 amino acids and includes an N-terminal FK506-binding protein (FKBP)-like domain which mediates interactions with the isoprenyl ligand of the PDE6 catalytic subunits [13,14,15] and a tetratricopeptide repeat (TPR) domain which is important for interacting with HSP70 and HSP90 [6,8] in addition to the inhibitory PDE6γ subunit [16]. Many *AIPL1* variations are associated with LCA4, commonly in the FKBP-like and TPR regions, affecting interactions with PDE6 and HSP90 [17].

Recent advances in in vitro retinal modelling have utilised organoids differentiated from induced pluripotent stem cells (iPSC) to recapitulate an approximation of the human retina, with well-defined inner nuclear layer (INL), outer plexiform layer (OPL), outer nuclear layer (ONL), external limiting membrane (ELM) and photoreceptor inner and outer segments [18,19]. These retinal organoids (ROs) contain populations of rod and cone photoreceptors throughout the ONL, with outer segment projections, which mediate presumptive phototransduction capabilities [18,19]. Notably, like human retinas, AIPL1 is localised in the ONL photoreceptor cells and inner segments of human RO, whereas the PDE6 subunits are detected within the outer segments [4,18,20,21].

ROs have been further adapted to enable the modelling of human retinal diseases and, to date, two published studies have sought to model LCA4 in vitro utilising this method. In these studies, LCA4 patient cells harbouring an *AIPL1* mutation were harvested from blood [20] or renal epithelial cells [21] and reprogrammed to iPSC. Five patient genotypes have been investigated, namely a c.256T>C, p.Cys89Arg homozygous mutation [22], and four patients carrying a c.834G>A, p.W278X mutation, either homozygously or heterozygously in trans with a c.466-1G>C splice mutation or c.665G>A, p.W222X nonsense mutation [21]. RO produced with these cells all retained a well-developed ONL with rod and cone photoreceptors while displaying a substantial reduction in AIPL1 protein and PDE6α [20], as well as a reduction in PDE6β and an increase in cGMP levels [21].

Thus far, each human in vitro model of LCA4 has relied on patient-derived cells with distinct AIPL1 mutations. These cells can be subject to ethical considerations and the cost prohibitive scalability may limit industrial and clinical applications. To address these limitations, we generated an AIPL1 CRISPR/Cas9 knockout model using commercially available human foreskin (BJ) fibroblasts to form isogenic retinal organoids. The AIPL1 knockout model faithfully recapitulates the disease-relevant molecular features evident in human patient models and thereby provides a complementary testing platform for potential LCA4 therapies [23].

## 2. Results

### 2.1. Generation of Isogenic AIPL1 Knockout iPSC by Simultaneous Reprogramming and CRISPR/Cas9 Gene Editing

To establish a human, isogenic photoreceptor model of AIPL1-related LCA, we set out to generate a genotype-matched isogenic iPSC line from a healthy individual using CRISPR/Cas9 technology to promote loss of AIPL1 expression (Figure 1). First, we designed CRISPR/Cas9 guide sequences to target the early exon 1 of *AIPL1* downstream of the starting codon (Appendix A). A Cas9-mediated double-strand break (DSB) in this region is expected to induce indel formation by the non-homologous end joining (NHEJ) repair pathway, resulting in the frameshift of the coding region and consequent gene silencing. Four gRNAs selected by in silico analysis—AIPL1 g1.1, g1.2, g1.3 and g1.4—were designed to direct the Cas9 endonuclease to *AIPL1* exon 1 (Appendix A). The targeting ability of each guide was tested in the HEK293T cell line transfected with AIPL1 gRNA and Cas9-encoding constructs PX458 (Appendix A). We observed that all guides effectively induced targeted indels in *AIPL1* exon 1 as detected by the T7 endonuclease I assay (Appendix A) with an approximate efficacy of 35% (Appendix A, *p* < 0.0001). We selected AIPL1 g1.4 for the generation of AIPL1 knockout (KO) iPSC by simultaneous reprogramming and CRISPR/Cas9 gene editing of healthy human BJ dermal fibroblasts [24] (Figure 1A). The emergence of BJ-derived iPSC colonies was detected as early as 14 days post-reprogramming, which were then isolated and expanded within 28 days (Figure 1B). To screen for AIPL1-edited iPSC, the genotype of single cell-derived iPSC clones was analysed by Sanger sequencing. Of a total of 66 clones, homozygous *AIPL1* KO was observed in 11 clones (16.7%), while heterozygous KO was observed in 3 clones (4.5%) and 52 clones (78.8%) were unedited (Figure 1C). We selected one iPSC clone with homozygous *AIPL1* KO through a 1 bp insertion at the AIPL1 g1.4 target (Figure 1D), resulting in a missense mutation and frameshift leading to a premature stop codon within exon 2 (p.H18Qfs*24). Immunocytochemistry analysis validated the pluripotency of both AIPL1 wild type (WT) and AIPL1 KO lines, with the detection of OCT4, TRA-1-60, NANOG and SSEA4 markers (Appendix A), and detection of trilineage potential toward germ layer differentiation (Appendix A). Finally, we validated the specificity of the AIPL1 KO by sequencing the 10 most likely off-targets predicted in silico with the highest homology to the AIPL1 g1.4 target sequence. No undesirable off-target mutations were detected in these lines (Appendix A).

### 2.2. Characterisation of Retinal Organoid Structure from AIPL1 KO iPSC

Next, we sought to evaluate the phenotypic effect of the CRIPSR/Cas9-introduced AIPL1 premature translation termination mutation. For this, AIPL1 WT and KO iPSC were differentiated into three-dimensional retinal organoids (RO) using a two-dimensional to a three-dimensional protocol for generating iPSC-derived neural retinal vesicles (NRV) [19]. In both cell lines, NRV was observed after 20–25 days of culture. Following mechanical isolation, NRV increased in diameter and developed clear lamination, and around day 170 began displaying brush borders comprising presumptive photoreceptor structures (Figure 2A). Evaluation of AIPL1 protein abundance found consistent expression in the outer nuclear layer of mature WT RO, but not in the AIPL1 KO model (Figure 2B). AIPL1 was detected in rhodopsin-positive rod photoreceptors (Figure 2C) and cone arrestin-positive cone photoreceptors (Figure 2D) as early as week 24 (day 170) in AIPL1 WT RO, but not in AIPL1 KO RO at any time point.

Additional RO characterisation found no observable differences between AIPL1 WT and KO RO for the expression of photoreceptor recoverin or photoreceptor cilia (ARL13B) during RO maturation (Appendix A), or in the expression of both L/M opsin and S opsin (Appendix A). Finally, ONL thickness (Figure 2E) and cell apoptosis (Figure 2F) were measured and quantified in the ONL of AIPL1 WT and KO RO with no significant difference found between model types, indicating loss of AIPL1 had minimal impact on RO development and maturation. This was confirmed at the transcriptional level, where comparable expression patterns were observed for markers of photoreceptor development (*CRX, NRL*) and function (recoverin (*RCVRN*), rhodopsin (*RHO*), cone opsins (*OPN1SW, OPN1MW/LW*) and phosphodiesterase subunits (*PDE6A, PDE6B, PDE6G, PDE6C, PDE6H*)) as well as markers of retinal cell types, specifically retinal ganglion cells (*BRN3B, ISLET1*), bipolar cells (*PKC-A*), Müller glia (*CRALBP*) and amacrine and horizontal cells (*PROX-1*) (Appendix A). Moreover, *AIPL1* transcription was retained in KO organoids, first detected in both WT and KO RO at 8 weeks with increased expression during RO maturation (Appendix A).

### 2.3. AIPL1 Knockout Results in RO with an LCA4 Phenotype

Downstream effects of AIPL1 protein loss were investigated in the AIPL1 WT and KO RO. Quantitative gene expression analysis was carried out for both AIPL1 and PDE6 subunits (Figure 3A), where expression was significantly increased over time, but no significant differences were measured between the WT and KO RO. Interestingly, a downward trend in AIPL1 expression was detected at each age examined in the AIPL1 KO compared to WT, but this did not reach significance. In addition, key proteins were measured in the whole RO for both WT and KO (Figure 3B). AIPL1 was consistently detected in WT RO but below the limit of detection in KO RO. Similarly, PDE6α was significantly reduced in KO RO (*p* = 0.032) compared to WT RO. Recoverin was used as a control for photoreceptor cells, and no difference was observed in recoverin levels between WT and KO RO, confirming that alterations in AIPL1 and PDE6α protein levels were unrelated to RO cellular makeup. The spatial location of rod-specific PDE6 enzymes (PDE6α and PDE6β) was also examined, where they were found to be abundant above the outer limiting membrane in WT but not in KO RO (Figure 3C). PDE6 enzymes function to hydrolyse cGMP within the retina, and the loss of these enzymes should lead to cGMP accumulation. When whole RO was quantified for cGMP concentrations, a significant increase (*p* = 0.041) was found in the AIPL1 KO model compared to WT (Figure 3D). Together, these data suggest AIPL1 KO RO manifest early molecular features of the LCA4 disease phenotype, including downstream effects on post-transcriptional levels of PDE6 and cGMP levels, and would be a suitable model to examine disease progression and treatments.

## 3. Discussion

LCA4 is an inherited retinal disease caused by loss-of-function mutations in AIPL1, leading to the loss of correctly assembled phosphodiesterase 6 (PDE6) enzymes, and subsequent lack of cGMP hydrolysis activity causing cGMP accumulation. Here, we generated an LCA4 model using CRISPR/Cas9 by introducing an early frameshift stop mutation leading to premature translation termination (p.H18Qfs*24).

RO generated from AIPL1 KO iPSC did not contain detectable levels of AIPL1 protein compared to RO from isogenic AIPL1 WT iPSC, and while transcript levels were generally lower in the KO model, this did not reach significance. The distance between the de novo terminal codon and the last exon–exon junction should have triggered the transcript degradation by nonsense-mediated decay (NMD), suggesting additional mechanisms rendering the AIPL1 transcript resistant to NMD [25]. Despite the lack of detectable AIPL1 protein, the AIPL1 KO RO model described here exhibited comparable morphology and development to AIPL1 WT RO and a similar abundance of rhodopsin-positive rods and opsin-positive cones within the outer nuclear layer. Previous in vitro models of LCA4 have exclusively utilised patient-derived iPSC to produce RO with LCA4-associated molecular features [20,21] and used CRISPR/Cas9 to correct the AIPL1 mutation to rescue these disease features [21]. These studies also determined that the LCA4 phenotype had no effect on photoreceptor cell development or relative abundance of other cell types within the retinal organoid during development, in line with the present study. In comparison, while no obvious morphological developmental changes were observed in the retina of *Aipl1*−/− mice, rapid photoreceptor degeneration coincident with outer segment elongation proceeded from postnatal day (P) 12 and was complete by 4 weeks [10]. Consistent with our AIPL1 KO model, reduced levels of the rod PDE6 subunits and increased cGMP levels were observed before the onset of degeneration [10]. While the temporal patterning of retinal progenitors and specification of the major retinal cell types are comparable in the developing human and mouse retina, the specification of all retinal cell types peaks before birth in the human developing retina whereas the specification of rod photoreceptors, bipolar cells and Müller glia peaks after birth in the mouse retina but before P8. Therefore, the reason why overt photoreceptor degeneration is not evident in the human LCA4 RO models is not immediately evident, but we hypothesise that the temporal window to observe photoreceptor degeneration in human RO is too short. Finally, in *Aipl1*−/− mice decreased expression of bipolar cell postsynaptic proteins at P8 prior to the onset of degeneration [26] suggest more nuanced dysregulation may be induced during early development prior to degeneration of rods and cones. Although not investigated within the scope of this study, it would certainly be an interesting avenue to explore bipolar cell specification and synaptic markers in human LCA4 RO in the future to better understand the early developmental LCA4 phenotype.

Importantly, AIPL1 KO was sufficient to significantly reduce PDE6α and PDE6β at the post-transcriptional level, accompanied by a significant increase in cGMP compared to isogenic WT RO which did not cause increased apoptosis in photoreceptors. These findings are in line with published patient-derived RO models of LCA4 [20,21] and in vivo *Aipl1*−/− mice [10] and imply an accurate and clinically relevant model has been generated in this study.

The model presented here recapitulates the molecular features of the LCA4 phenotype seen in other model types and therefore provides an alternative, complementary, model system to further interrogate the early molecular pathways involved in disease pathogenesis. Moreover, we anticipate that the model described here will be useful for testing potential gene therapies to treat LCA4 in a mutation-independent manner. One such option is adeno-associated virus (AAV) mediated gene therapy, which is currently a popular choice due to the versatility, good safety profile, and low immunogenicity reported [27,28]. AAV-RPE65 treatment was granted FDA approval as the first gene therapy for an inherited retinal degenerative disease for administration in patients with LCA2 [29,30]. AAV treatment of AIPL1 deficient mice shows good recovery by using CMV [31,32], RK [33], or RHO [32] as a promotor to drive AIPL1 expression and either AAV2/2 [31], AAV2/5 [33], or AAV2/8 [31,32,33] serotypes. Furthermore, the efficacy of different AAV serotypes has been measured in RO, with AAV7m8 [34], and AAV2/2 [35] shown to produce the strongest transduction in each respective study. Finally, AAV2/5-RP2 has previously been tested in retinal organoids, where it was able to restore a healthy phenotype [36], displaying the utility of RO to examine AAV-mediated rescue in a human in vitro system.

Another prospective use of this model results from the elevated cGMP observed, which is a shared downstream pathway in many IRD [37] with high levels of cGMP and cGMP-dependent protein kinase (PKG) linked to rod and cone cell death [38]. Recent work has shown that cGMP analogues can act as a cGMP inhibitor to rescue photoreceptor degeneration in vivo and in murine-derived cell cultures [39,40]. Additionally, downstream molecular pathways shared in IRD and related to elevated cGMP remain viable targets for the development of future therapeutics [41], including PKG [42], PARP activity [42,43] and Ca^2+^-dependent calpain activity [44]. While investigating the activation of these pathways was outside the scope of this study, it is interesting to note that cGMP-mediated PR degeneration was not observed at any time in this model during differentiation (230 days in culture), suggesting that PR degeneration may proceed at a later time point and highlighting the required duration of RO culture as a limitation of this model system. However, the observed increased cGMP demonstrates the value of this model as a highly beneficial intermediary for testing novel therapeutics targeting the elevated cGMP levels and downstream dysregulated pathways in IRD to gain meaningful data from a human-derived in vitro model prior to proceeding towards clinical trials.

## 4. Materials and Methods

### 4.1. Cell Culture

Human embryonic kidney 293T (HEK293T) cells (American Type Culture Collection; ATCC, Manassas, VA, EUA) were cultured in Dulbecco’s modified Eagle’s medium (DMEM; Life Technologies, Carlsbad, CA, USA) supplemented with 10% (*v*/*v*) foetal bovine serum (FBS; Gibco, Loughborough, UK), 2 mM L-glutamine and 1% (*v*/*v*) penicillin–streptomycin (Gibco). Cells were routinely passaged using Trypsin 0.05%. Induced pluripotent stem cells (iPSC) were maintained on Geltrex-coated 6 well plates in mTeSR Plus media (Stemcell, Cambridge, UK). Cells were routinely passaged using Versene (Gibco). Cells were maintained at 37 °C in a humidified atmosphere of 5% CO_2_.

### 4.2. Design of CRISPR/Cas9 Sequences for AIPL1 Knockout

CRISPR/Cas9 guide RNAs (gRNAs) were designed in silico to target the AIPL1 exon 1 using Benchling CRISPR design software (https://www.benchling.com). The top four gRNA candidates (Appendix A) with optimal on-target efficacy and specificity were cloned into BbsI sites of pSpCas9(BB)-2A-GFP (PX458, Addgene plasmid #48138) vector encoding CBh-driven Cas9 nuclease and U6-driven gRNA. To validate AIPL1-targeted CRISPR/Cas9 sequences, HEK293T cells were transfected with PX458 encoding AIPL1-targeting gRNAs by Lipofectamine 2000 (Thermo Fisher Scientific, Loughborough, UK) according to manufacturer instructions. CRISPR/Cas9-mediated indel formation at the AIPL1 target sequence was assessed by T7 endonuclease I (T7EI) assay for detection of DNA heteroduplexes according to Ran et al. 2013 [45]. Primers used for cloning AIPL1 gRNAs and for amplification of AIPL1 exon 1 for T7EI assay are represented in Appendix A.

### 4.3. Generation of AIPL1 Knockout Isogenic iPSC

AIPL1 KO isogenic iPSC were generated from control BJ dermal fibroblast line (ATCC CRL-2522) by simultaneous reprogramming and CRISPR/Cas9 knockout according to Howden et al. 2015 [24]. Briefly, BJ fibroblasts were nucleofected with reprogramming plasmids—pCXLE-hOct3/4-shp53-F (Addgene plasmid #27077), pCXLE-hSK (Addgene plasmid # 27078), pCXLE-hUL (Addgene plasmid #27080) and pSimple miR302/367 (Addgene plasmid #98748)—and PX458 constructs encoding AIPL1-targeted CRISPR/Cas9 sequences. Single-cell derived iPSC colonies were manually picked at approximately 25–28 days post-nucleofection and expanded for genotyping analysis. Genomic DNA from each iPSC clone was extracted using the QuickExtract reagent (Lucigen, Middleton, WI, USA) solution and screening of AIPL1 KO clones was performed by Sanger sequencing. Off-target analysis was performed by in silico prediction of the top 10 regions with highest homology to the AIPL1-targeting sgRNA using Benchling software (https://www.benchling.com) and Sanger sequencing to evaluate the presence of undesirable mutations caused by CRISPR/Cas9 off-target activity. Retention of iPSC potential for trilineage differentiation following CRISPR/Cas9 treatment was confirmed using the STEMdiff™ Trilineage Differentiation Kit (Stemcell Technologies, Cambridge, UK) as per manufacturer’s instructions with RNA assessed for *Oct4* (pluripotency), *Pax6, Nestin* and *Otx2* (Ectoderm), *T* and *Cdx2* (Mesoderm), and *AFP*, *Sox17* and *Gata6* (Endoderm) to confirm differentiation to three germ lineages.

### 4.4. Generation of Retinal Organoids (RO)

Directed differentiation of iPSCs into 3D RO was carried out as previously described [19,21]. iPSCs were dissociated with Versene (Gibco, Loughborough, UK), clumps were collected, washed twice with PBS, resuspended in culture medium, and seeded onto Geltrex (Gibco) coated 6-well plates. iPSC colonies were grown until 90–95% confluent, at which point they were cultured in Essential 6TM media (Gibco) for 2 days (Day 1 and 2 of differentiation) then neural induction media (Advanced DMEM/F12 (1:1), 1% non-essential amino acids (NEAA), 1% N2 supplement, 1% GlutaMAX, 1% antibiotic–antimycotic (All Gibco)) until development of neuro-retinal vesicles (NRVs), typically between day 28 and 42. NRVs were manually excised using 21G needles/scalpel blades and grown in low-binding 96-well plates (96-well Nunc Sphera Round Bottom Plates, Thermo Fisher Scientific) in retinal differentiation media (DMEM/F12 nutrient mix (3:1 ratio; Gibco), 10% foetal bovine serum (FBS; Gibco), 2% B27 supplement (without vitamin A; Gibco), 100 μM taurine (Tocris, Abingdon, UK), 2 mM GlutaMAX (Gibco, Loughborough, UK) and 100 U/mL penicillin/streptomycin (Gibco, Loughborough, UK)), with media changes every 2 days. At day 70, cultures were supplemented with 1 μM retinoic acid, and ROs were transferred into low-binding 24-well plates. At day 84, the cultures were supplemented with 1% N2 and RA lowered to 0.5 μM. At day 100, B27 and RA were removed from the medium.

### 4.5. Immunofluorescence and Imaging

ROs were fixed in 4% PFA, 5% sucrose in PBS for 30 min at 4 °C then dehydrated for 1 h each in 6.25%, 12.5% and 25% sucrose:PBS at 4 °C, ROs were embedded in Tissue-Tek OCT compound and stored at −80 °C until sectioned. The amount of 7 µm cryosections was mounted on SuperFrost PlusTM slides (Thermo Fisher Scientific) and stored at −20 °C. Slides were stained for ICC by first incubating in block buffer (10% donkey serum (Sigma-Aldrich, Gillingham, UK), 0.01% Triton-X (Sigma-Aldrich, Gillingham, UK) in PBS) for 1 hr at room temperature (RT), then incubating with a primary antibody for 1 h (RT; antibody-specific dilutions; Appendix A). Slides were washed 3 times with PBS and incubated with species-specific secondary antibodies for 45 min (1:1000; Appendix A). If required, slides were then incubated with stains (Appendix A) for 20 min before further incubation with 4,6-diamidino-2-phenylindole (DAPI; 2 mg/mL) (Invitrogen, Loughborough, UK) in PBS for 5 min, then washed 3 times with PBS. Slides were dried at RT and mounted in fluorescence mounting media (Dako; Agilent, CA, USA). Slides used for TUNEL assay (In Situ Cell Death Detection Kit; Merck) were stained as per kit guidelines, then counterstained with DAPI and mounted as per the above protocol. All images were acquired using a Leica Stellaris 5 laser-scanning confocal microscope. Images were prepared using Adobe Photoshop, Image J and Adobe Illustrator CS6.

### 4.6. Image Quantification

ONL thickness was calculated using ImageJ by taking 10 unique measurements from 3 DAPI-stained RO. TUNEL positive cells were quantified using low magnification images where ONL length was calculated using ImageJ and positive cells were manually counted. Statistical significance was determined by unpaired Student’s *t* test, where * denotes a *p* value < 0.05.

### 4.7. RNA Extraction and Quantitative PCR (qPCR)

RNA from iPSC and ROs was extracted using the PicoPure RNA Isolation Kit (Thermo Fisher Scientific). cDNA synthesis was performed using the High-Capacity cDNA Reverse Transcription Kit (Thermo Fisher Scientific). 2× GoTaq Green master mix (Promega, Southampton, UK) was used for DNA amplification by PCR with standard cycling conditions for semi-quantitative PCRs. Real-time PCR reactions were set up with 2× LabTaq Green Hi Rox Master Mix (Labtech, East Sussex, UK) and validated primers (0.25 pmol/uL; sequences detailed in Appendix A) and run on an Applied Biosystems QuantStudioTM 6 Flex real-time PCR system. Gene expression levels were normalised to CRX as a consistently expressed PR-specific reference gene to negate differences in size and cellular makeup between samples. Three biological replicates per sample were used to calculate averages and standard deviation, and statistical significance was determined by one-way ANOVA where *, **, *** denote a *p* value ≤ 0.05, ≤0.01 and ≤0.005, respectively.

### 4.8. Western Blotting

Protein was isolated from 3–6 pooled RO by first incubating in RIPA buffer for 15 min on ice, briefly sonicating, then centrifuging at 4 °C for 10 min at 13 k rpm. Protein concentration was quantified by Pierce BCA assay (Thermo Fisher Scientific) as per manufacturer’s instructions. Protein concentrations were normalised by addition of RIPA buffer, then mixed 1:1 with 2× SDS PAGE buffer and heated at 95 °C for 5 min to facilitate protein denaturation. Samples were run on precast 4–20% gradient gels (Bio-Rad, Watford, UK) at 120 V for 2 h, or until the dye reached the end of the gel. The proteins were transferred to a nitrocellulose membrane by wet transfer, at 90 V for 1.5 h, using standard protocols. Membranes were blocked for 1 h at room temperature in 5% skim milk powder in PBS-T, then incubated overnight at 4 °C in primary antibody (Appendix A). The following day membranes were incubated in species-specific HRP-conjugated secondary antibody for 1 h, then developed using Clarity MAX Western ECL (Bio-Rad, Watford, UK) for 5 min. Membranes were imaged using a ChemiDoc MP Imaging System (Bio-Rad, Watford, UK). Images were formatted using Image Lab (Bio-Rad, Watford, UK), which was also used to calculate band density. Three independent protein isolates were used per condition and statistical significance was calculated by unpaired Students *t* test, where * denotes a *p* value < 0.05.

### 4.9. Quantification of cGMP

cGMP concentrations were calculated for individual whole RO by ELISA (Cambridge Biosciences, Cambridge, UK) according to the manufacturer’s instructions. In brief, RO were washed with PBS and then incubated in 100 μL of 0.1 M HCl for 20 min at RT and mechanically homogenised. The samples were centrifuged at 1000× *g* for 10 min then the supernatant was combined with 200 μL of ELISA buffer. An amount of 50 μL of this sample was used per well. Absorbance was measured at a wavelength of 420 nm. Values given are normalised to volume to give cGMP concentration per the whole RO, and statistical significance is calculated by unpaired Students *t* test, where * denotes a *p* value < 0.05.

## Figures and Tables

**Figure 1 ijms-24-05912-f001:**
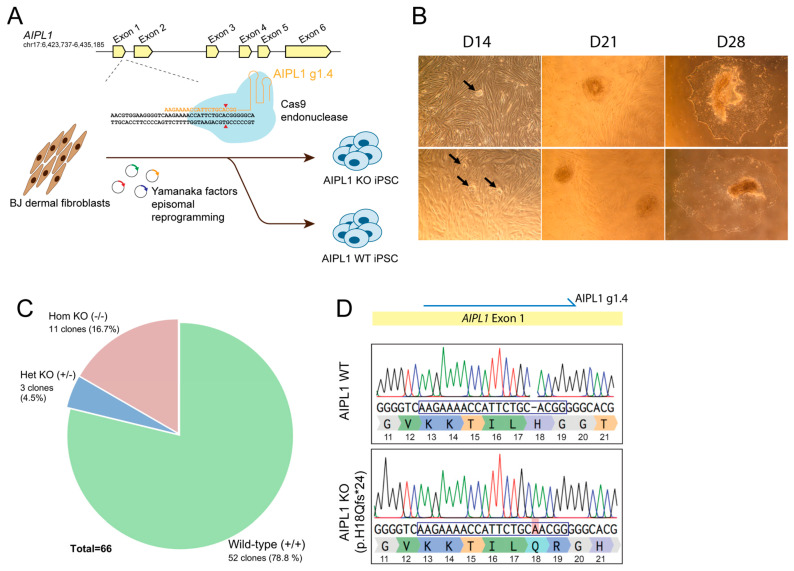
Generation of AIPL1 knockout iPSC isogenic lines to model AIPL1-related Leber congenital amaurosis. (**A**) Schematic illustration of simultaneous iPSC reprogramming and CRISPR/Cas9-mediated AIPL1 knockout. BJ dermal fibroblasts from a healthy individual were used as the cell source and nucleofected with reprogramming constructs encoding Yamanaka factors, along with a CRISPR/Cas9 endonuclease directed by a guide RNA (AIPL1 g1.4) to exon 1 of the AIPL1 gene. Red arrowhead, Cas9 cleavage site. Edited AIPL1 isogenic lines were isolated and compared to non-edited AIPL1 wild-type lines for modelling AIPL1 LCA. (**B**) Brightfield images of iPSC colonies at 14, 21 and 28 days post-reprogramming. Emerging colonies at 14 days are indicated by black arrows. Scale bar = 10 µm. (**C**) Genotype of AIPL1 knockout in iPSC clones. Single cell-derived iPSC clones isolated from the reprogrammed population were genotyped by Sanger sequencing. The distribution of AIPL1 non-edited wild-type clones (+/+) and clones with heterozygous (+/−) or homozygous (−/−) AIPL1 knockout is shown. (**D**) Sequence alignment of a selected AIPL1 knockout clone against isogenic non-edited AIPL1. The AIPL1 g1.4 sequence is indicated by a blue box. Homozygous insertion of one adenine nucleotide–highlighted in red–in the AIPL1 knockout line results in frameshift disruption and formation of a premature stop codon (p.H18Qfs*24).

**Figure 2 ijms-24-05912-f002:**
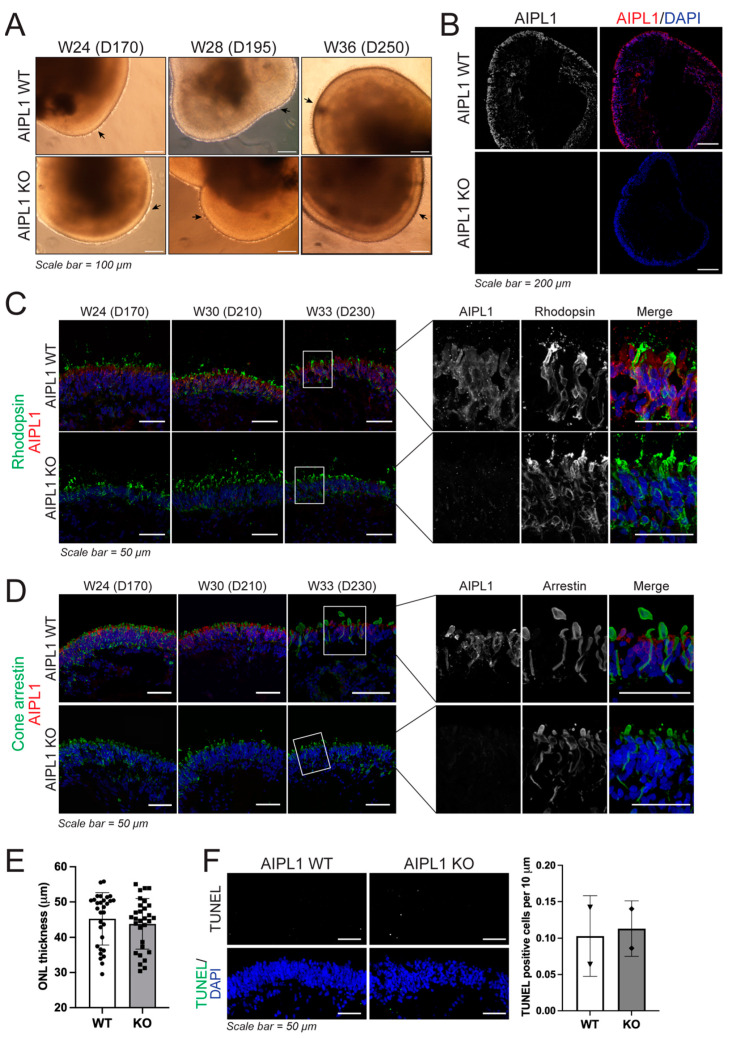
Isogenic AIPL1 WT and KO iPSC generate retinal organoids (RO) with comparable structures during differentiation. (**A**) Brightfield images of developing AIPL1 WT and AIPL1 KO RO (week (W) 24–36, day (D) 170–250). A well-developed brush border of presumptive OS/IS photoreceptor structures is seen on the surface of RO from D170 (arrow). Scale bars = 100 µm. (**B**) AIPL1 KO RO (D230) had no detectable AIPL1 protein (red) compared to AIPL1 WT RO. Scale bars = 200 µm. (**C**) The development of rhodopsin-positive rod photoreceptors (green) was comparable in AIPL1 WT and AIPL1 KO RO at D170 (W24), D210 (W30) and D230 (W33), despite the loss of AIPL1 protein (red) in AIPL1 KO RO. Scale bars = 50 µm. The white rectangles demarcate the highlighted regions in the panel to the right. (**D**) The development of cone arrestin-positive cone photoreceptors (green) was comparable in AIPL1 WT and AIPL1 KO RO at D170 (W24), D210 (W30) and D230 (W33), despite the loss of AIPL1 protein (red) in AIPL1 KO RO. Scale bars = 50 µm. The white rectangles demarcate the highlighted regions in the panel to the right. (**E**) ONL thickness was unaltered between AIPL1 WT and KO RO (W33, D230). Each dot represents an individual measurement, shown as mean +/− SD with significance determined by unpaired *t* test. (**F**) Apoptotic cells detected by TUNEL assay and number of positive cells per 10 µm of RO ONL. A total of 2 RO were included per timepoint and condition, shown as mean +/− SD with significance determined by unpaired *t* test.

**Figure 3 ijms-24-05912-f003:**
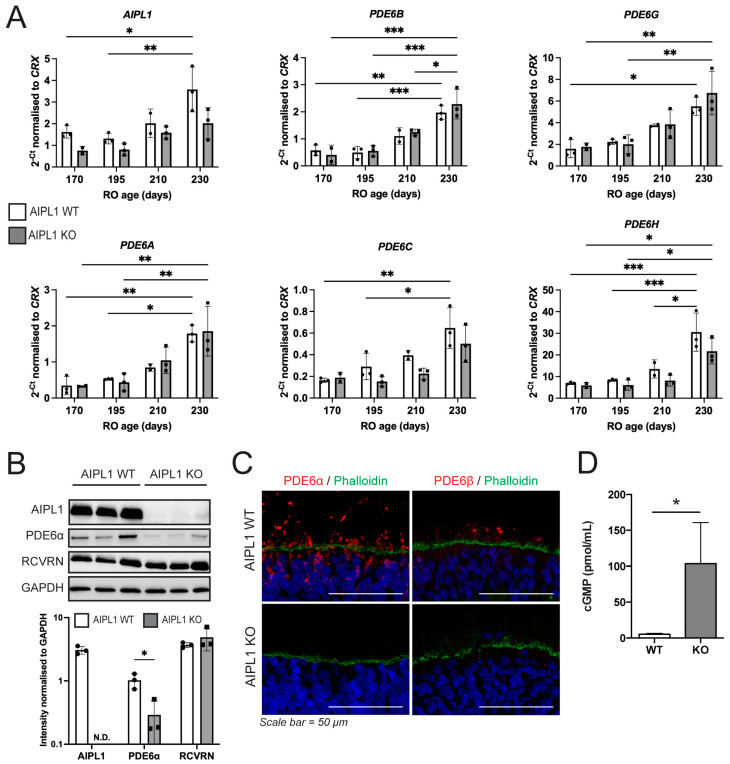
Retinal organoids derived from AIPL1 KO iPSC mimic early molecular features of LCA4 with no detectable PDE6 protein and elevated levels of cGMP. (**A**) qPCR of mature RO (W24-W33, D170-230) found minimal differences between AIPL1 WT and AIPL1 KO RO. RT-PCR for W8-W33 pooled samples is shown in Appendix A. Three biological replicates; mean +/− SD. Significance calculated by 2-way ANOVA where *, **, *** indicates *p* < 0.05, 0.01 and 0.005, respectively. (**B**) Western blot of 25 µg protein isolated from 3–6 pooled mature RO (W33, D230) for AIPL1, PDE6α and recoverin (RCVRN), with GAPDH included as a loading control. Bands were quantified and normalised to GAPDH to show relative band intensity. Each lane contained protein from separate isolates. Three biological replicates; mean +/− SD. Significance calculated by unpaired *t* test where * indicates *p* < 0.05. N.D. Not detected. (**C**) IF analysis of rod cGMP PDE6α and PDE6β (red) abundance in the ONL region of W30 (D210) RO identified no positive staining in AIPL1 KO RO. Scale bars = 50 μm. (**D**) cGMP ELISA analysis of whole W33 (D230) RO showed elevated cGMP in AIPL1 KO RO compared to WT. Three biological replicates; mean +/− SD. Significance calculated by unpaired *t* test where * indicates *p* < 0.05.

## Data Availability

The data presented in this study are contained within the article and Appendix A.

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
