# Peer review of "Retinal Organoids from an AIPL1 CRISPR/Cas9 Knockout Cell Line Successfully Recapitulate the Molecular Features of LCA4 Disease"

_ijms, 2023, doi:10.3390/ijms24065912_

Round 1

Reviewer 1 Report

This manuscript describes the generation and characterisation of iPSC-retinal organoid lacking AIPL1. This is an excellent approach to model LCA caused by mutations in AIPL1. It complements the existing mutant organoids (some generated by the same authors) and animal models. Given the existence of these models and the advances in the field, I have a few concerns/suggestions.

1. Authors should investigate the levels of cone PDE6 and cone proteins similar to their analysis of rod proteins in Fig. 3B.

2. Unlike mouse models, photoreceptors do not degenerate in organoid models. Authors should discuss the reasons behind the observed difference.

3. Is AIPL1 localised in the COS, as shown in Fig. 2D?

3. While not necessary, given the advances in the field, it is surprising that the ultrastructural details on OS and their function are not attempted. 

Minor

Panel 2D, should be D170

Author Response

We would like to sincerely than the reviewer for their time, and constructive review and suggestions for the manuscript. 

1. Authors should investigate the levels of cone PDE6 and cone proteins similar to their analysis of rod proteins in Fig. 3B.

Response: Unfortunately, we have tried a range of commercially available antibodies for the cone specific PDE6 subunits, resulting in high non-specific background in human retinal organoids by IHC and undetectable/non-specific bands by WB. We have contacted researchers in the field to request in-house developed and verified antibodies which will hopefully overcome this problem in our future work.

2. Unlike mouse models, photoreceptors do not degenerate in organoid models. Authors should discuss the reasons behind the observed difference.

Response: We have expanded our discussion accordingly (lines 298-310) to address the possible reasons for this difference. 

3. Is AIPL1 localised in the COS, as shown in Fig. 2D?

Response: We are very grateful to the reviewer for bringing this to our attention. AIPL1 does not localise in COS, as seen in W24 (D170) and W30 (D210) RO, or as seen in human and mouse retina by immunohistochemistry. The apparent co-localisation in Fig. 2D is a technical issue due to poor RO section selection (where we note the extrusion of nuclei in the 'COS' shown), and we have therefore replaced this image in the figure. We sincerely apologise for this and are pleased we have been able to rectify the issue. 

4. While not necessary, given the advances in the field, it is surprising that the ultrastructural details on OS and their function are not attempted.

Response: We appreciate this suggestion and is certainly an aspect we will address in our future work. We showed using ultrastructural analysis that LCA4 patient retinal organoids formed clear connecting cilia with stacked discs in the proximal part of the outer segments (Leung et al 2022) and is something we wish to investigate further. It is notable, however, that the formation of extended outer segments with characteristic stacked discs has been something of a technical challenge in the field given the lack of morphological support for the outer segments, but we have been refining our protocols to improve this. 

Panel 2D, should be D170: Thank you for bringing this error to our attention. We have now corrected this. 

Reviewer 2 Report

This manuscript is well done. The writing is clear and concise. The experiments showed creation of the iPSCs and retinal organoid development with and without the mutation. There are not many changes that I would make. I really appreciate such a well-written manuscript. I only have a few minor comments.

You mention several times that you have made a LCA4 model that recapitulates the disease (in the title and throughout the manuscript). I think you only showed that knock out of AIPL1 results in an increase in cGMP (which is a nice measurable downstream affect). I think it may be too bold to say that these ROs recapitulate disease since LCA4 is characterized by PR degeneration, electrophysiological defects, and features associated with retinitis pigmentosa (bone spicules, disc pallor, and arteriole attenuation). You should consider revising to indicate a physiological phenotype similar to what may be happening early in LCA4 human patients.  

Please define “BJ”. Line 76

Lines 226-231. You suggest that synapse formation between photoreceptors and bipolar cells (BPCs) may be required prior to active cell death of PRs in AIPL1-LCA. You checked RNA for BPCs, but did you look for BPC protein markers by IF? In my experience, BPCs are very late in development using ROs so your suggested timing of cell death would explain the fact that your ROs had very few if any BPCs (and hence no cell death differences).

Figure S4: please indicate whether the numbers at the top of the blots are days or weeks.

[comment for future work. Have you thought about checking whether the PDE6A, AIPL1, and cGMP concentrations can be restored with AAV2/8-RHO(or CMV)-hAIPL1?]

Author Response

We would like to sincerely thank the reviewer for taking the time to review our manuscript. We are grateful for the insightful and constructive comments that have improved the manuscript.

Reviewer: You mention several times that you have made a LCA4 model that recapitulates the disease (in the title and throughout the manuscript). I think you only showed that knock out of AIPL1 results in an increase in cGMP (which is a nice measurable downstream affect). I think it may be too bold to say that these ROs recapitulate disease since LCA4 is characterized by PR degeneration, electrophysiological defects, and features associated with retinitis pigmentosa (bone spicules, disc pallor, and arteriole attenuation). You should consider revising to indicate a physiological phenotype similar to what may be happening early in LCA4 human patients.  

Response: We appreciate this comment and have rephrased our manuscript where appropriate to emphasise that our model manifests early molecular changes associated with the loss of AIPL1. 

Reviewer: Please define “BJ”. Line 76

Response: We have revised our text to indicate that these are commercially available human foreskin fibroblasts. 

Reviewer: Lines 226-231. You suggest that synapse formation between photoreceptors and bipolar cells (BPCs) may be required prior to active cell death of PRs in AIPL1-LCA. You checked RNA for BPCs, but did you look for BPC protein markers by IF? In my experience, BPCs are very late in development using ROs so your suggested timing of cell death would explain the fact that your ROs had very few if any BPCs (and hence no cell death differences).

Response: This is an excellent point and suggestion raised by the reviewer. We plan to investigate bipolar cell and synapse markers in our future work by immunofluorescence and other technologies (e.g. RNA sequencing), to explore this hypothesis and gain better insights into the early molecular changes associated with the loss of AIPL1 in human retinal organoids.

Reviewer: Figure S4: please indicate whether the numbers at the top of the blots are days or weeks.

Response: We thank the reviewer for spotting this omission and we have updated Figure S4 accordingly.

Reviewer: [comment for future work. Have you thought about checking whether the PDE6A, AIPL1, and cGMP concentrations can be restored with AAV2/8-RHO(or CMV)-hAIPL1?]

Response: We have indeed thought of this and are currently preparing a manuscript which excitingly shows the rescue of AIPL1, PDE6 and cGMP by AAV gene therapy using both our AIPL1 knockout and patient retinal organoid models. 

Round 2

Reviewer 1 Report

The authors have addressed my concerns. Congratulations on the nice work.